# Differential Protein Expression in Extracellular Vesicles Defines Treatment Responders and Non-Responders in Multiple Sclerosis

**DOI:** 10.3390/ijms251910761

**Published:** 2024-10-06

**Authors:** Gabriel Torres Iglesias, MariPaz López-Molina, Lucía Botella, Fernando Laso-García, Beatriz Chamorro, Mireya Fernández-Fournier, Inmaculada Puertas, Susana B. Bravo, Elisa Alonso-López, Exuperio Díez-Tejedor, María Gutiérrez-Fernández, Laura Otero-Ortega

**Affiliations:** 1Neurological Sciences and Cerebrovascular Research Laboratory, Department of Neurology, Neurology and Cerebrovascular Disease Group, Neuroscience Area of Hospital La Paz Institute for Health Research—IdiPAZ (La Paz University Hospital—Universidad Autónoma de Madrid), 28046 Madrid, Spain; gabri_t13@hotmail.com (G.T.I.); mplm1995@gmail.com (M.L.-M.); luciabotella733@gmail.com (L.B.); fernilaso.9@gmail.com (F.L.-G.); beatriz.lapaz2@gmail.com (B.C.); fernandezfournier@hotmail.com (M.F.-F.); inmapuertas@hotmail.com (I.P.); elisaalonso164@hotmail.com (E.A.-L.); exuperio.diez@salud.madrid.org (E.D.-T.); mariagutierrezfdez@hotmail.com (M.G.-F.); 2Proteomics Unit, Health Research Institute of Santiago de Compostela (IDIS), 15706 Santiago de Compostela, Spain; sbbravo@gmail.com

**Keywords:** biomarkers, extracellular vesicles, multiple sclerosis, protein profiling, treatment response

## Abstract

Multiple sclerosis (MS) remains the leading cause of neurological disability among young adults worldwide, underscoring the urgent need to define the best therapeutic strategy. Recent advances in proteomics have deepened our understanding of treatment mechanisms and revealed promising biomarkers for predicting therapeutic outcomes. This study focuses on the identification of a protein profile of circulating extracellular vesicles (EVs) derived from neurons, oligodendrocytes, and B and T cells able to differentiate treatment responders and non-responders in 80 patients with MS. In the patients who responded to treatment, T cell-derived EVs were enriched in LV151, a protein involved in the promotion of anti-inflammatory cytokines, whereas Bcell-derived EVs showed elevated PSMD6 and PTPRC, related to immunoproteasome function. Oligodendrocyte- and neuron-derived EVs showed upregulated CO6A1 and COEA1, involved in extracellular matrix reorganisation, as well as LAMA5, NonO, SPNT, and NCAM, which are critical for brain repair. In contrast, non-responders showed higher levels of PSMD7 and PRS10 from B cell-derived EVs, associated with DNA damage, and increased levels of PERM and PERL from T cell-derived EVs, linked to nuclear factor kappa B activation and drug-resistant proteins such as HS90A and RASK. These findings highlight a distinct panel of proteins in EVs that could serve as an early indicator of treatment efficacy in MS.

## 1. Introduction

Multiple sclerosis (MS) is an inflammatory, demyelinating, neurodegenerative disease of the central nervous system (CNS) that affects approximately 2.8 million people worldwide. It is more prevalent in women, with a female-to-male ratio of 3:1, and it typically manifests in young adults between the ages of 20 and 45 [1]. Despite significant advancements in the diagnosis and treatment of MS in recent years, the disease remains the leading cause of non-traumatic neurological disability, making patients highly vulnerable and sometimes dependent [1]. It is essential they receive better treatments to reduce the disease’s activity as well as its associated disability, thereby improving their quality of life. To ensure that effective treatment for MS is possible, 20 specific disease-modifying treatments (DMTs) have been developed in the last 2 decades. These therapies are based on modulating the activity of the immune system to minimise its aggressive response to the CNS [2]. However, despite these advances in treatment development, more than 30% of patients do not respond effectively to the chosen DMT, having to switch to another therapy whose ultimate effectiveness is unknown. To date, there is no biomarker for selecting the most appropriate treatment in clinical practice, and current therapeutic monitoring is based on waiting for the appearance of a new clinical relapse or new lesions in the CNS, which puts the patient’s health at risk [3]. Addressing this issue could be significantly aided by identifying an effective biomarker that can rapidly indicate the likelihood of therapeutic failure.

In this regard, an ideal biomarker should be easily obtainable through minimally invasive or non-invasive methods and be reproducible, highly sensitive, and specific while remaining cost-effective [4]. However, the CNS presents unique challenges due to its structural and functional isolation, necessitating highly invasive procedures for direct sample acquisition. Additionally, potential molecular markers in commonly accessible samples such as blood are often present in very low concentrations, making their detection and quantification difficult and expensive [5].

Currently, oligoclonal bands are the primary laboratory biomarker used in clinical practice; however, their utility is limited because they are obtained from cerebrospinal fluid. Neurofilament light chains are also being used to monitor the response to DMTs in MS. However, elevated levels of this biomarker indicate non-specific axonal damage that can be triggered by various causes [6]. Identifying a laboratory biomarker that is more closely related to the disease pathogenesis would enhance specificity and deepen our understanding of MS.

Advancements in -omics technology in recent years, especially since the early 2000s, have led to a better understanding of disease pathogenesis [7]. Genomics has been the most extensively studied. Despite these advancements, genomics alone has not fully elucidated the complexity of the disease [5]. It has become clear that the products of gene expression are often more intricate and closely tied to specific phenotypic traits than to the genes themselves [5]. This realisation has led to the emergence of various -omics fields, each focusing on different biological aspects. Proteomics, which analyses the entire protein content, function, and interaction in a cell or organism, has proven superior for biomarker research due to its focus on proteins, the active molecules in biological processes [8,9].

Recent progress in the study of extracellular vesicles (EVs) positions them as potential biomarkers in several medical fields, particularly for diseases affecting the nervous system. EVs are clusters of nano-sized membrane particles, varying in size, cargo, and surface markers, secreted into the extracellular environment. They carry cytoplasmic and cell membrane components, including RNA, metabolites, and proteins [10]. The membranous envelope of EVs protects their contents from degradation [11]. Secreted by all living cells, EVs play crucial roles in physiological functions and pathological processes, mediating intercellular communication. Several studies have shown that EV content, including proteins [12], provides valuable insights into the pathophysiology of MS and could serve as accessible biomarkers for disease monitoring. In this regard, EVs have been associated with relapses and the development of new white matter lesions [13]. Moreover, several studies have indicated that DMTs can specifically modify EV content depending on the treatment type [14,15], although this area requires further exploration.

Classifying EVs according to their cellular origin is important to specifically understand the function of each cell type in the pathogenesis of the disease [16]. Brain cells release EVs containing information about the brain damage processes occurring in MS. Due to their small size and membrane composition, these EVs from the nervous system can cross the blood brain barrier (BBB) and pass into the bloodstream. Thus, circulating EVs serve as an accessible source of CNS biomarkers, providing insight into the pathological processes of MS [17], bypassing the need for invasive methods to obtain information from the CNS. Additionally, immune cells involved in the inflammatory response release EVs into the bloodstream, which could offer relevant information about the immune system’s pathological reactions in MS. Therefore, nervous and immune-derived EVs comprise a mixture of vesicles from various cell types, containing more complete data on the pathological processes of the disease [18].

In this study, we analysed the proteomics of EVs from T cells, B cells, neurons, and oligodendrocytes in patients with MS initiating a new DMT, with the aim of analysing whether the protein content of EVs could provide relevant information on the biological processes that explain the response or failure of treatment. This approach aims to identify early potential responders and non-responders, guiding the selection of the most appropriate treatment for each patient based on their proteomics profile. Understanding the biological basis of these mechanisms paves the way towards the discovery of therapeutic strategies for personalised medicine.

## 2. Results

We enrolled 80 patients with remittent recurrent MS (RRMS) starting a new DMT. Forty-nine patients were considered responders, and 31 were non-responders (Figure 1). Patient characteristics are summarised in Table 1.

### 2.1. Characterisation of Brain- and Immune System-Derived Extracellular Vesicles: Size, Morphology, and Marker Profiling

Brain- and immune system-derived EVs were characterised for size, morphology, and presence of specific EV marker profiles in their membrane (Figure 2). EV samples showed the typical morphology of lipid bilayer spheres, with a size of approximately 200 nm by transmission electron microscopy and less than 300 nm by nanoparticle tracking analysis (NTA). Triple positivity for the EV-specific tetraspanins CD9, CD63, and CD81 allowed for a robust characterisation of the EV samples by Western Blot.

### 2.2. Immune System-Derived Extracellular Vesicles Are Involved in Immune Modulation in Responder Patients

Upon analysing the protein content in PRE- and POST-treatment assessments, we identified 270 differentially expressed proteinsbetween responders and non-responders. These proteins are represented in the volcano plots of Figure 3 illustrating their cell origin from T cell- (A), B cell- (B), neuron- (C), and oligodendrocyte-derived EVs (D). These proteins are also included in Appendix A.

Among the 270 differentially expressed proteins, 133 were enriched in the responder patients. To determine key mechanisms underlying treatment response, the most significant proteins associated with specific pathogenic pathways were selected using Reactome software.

Before treatment (PRE), T cell-derived EVs in responders showed upregulation of the LV151 protein compared with non-responders (40,806.66 ± 4868.62 and 15,517.37 ± 16,522.99 normalised spectral abundance factor [NSAF]; *p* = 0.0018; fold change [FC] = 0.38) (Figure 4A). This protein is involved in the synthesis of the anti-inflammatory cytokines interleukin (IL)-4 and IL-10 (Figure 5B). Three months after treatment onset (POST), PSMD6 was overexpressed in B cell-derived EVs from responders compared with non-responders (8102.59 ± 2548.93 and 1226.93 ± 668.68 NSAF; *p* = 0.01; FC = 0.15) (Figure 4A). This protein has been shown to be enriched for functions related to antigen processing and cross-presentation (Figure 5B). Furthermore, at 3 months POST-treatment, PTPRC was upregulated in B cell-derived EVs from responders (9156.95 ± 1660.55 and 2694.83 ± 1588.30 NSAF; *p* = 0.008; FC = 0.29) (Figure 4A), playing a role in antibody-dependent immune responses.

### 2.3. Nervous System-Derived Extracellular Vesicle Subpopulation Has a Role in the Extracellular Matrix Reorganisation and White Matter Repair in Responder Patients

Before treatment, EVs from oligodendrocytes overexpressed the CO6A1 protein before treatment compared with 3 months POST-treatment in responder patients (123,425.52 ± 23,160.89 and 49.66 ± 18,082.41 NSAF; *p* = 0.01; FC = 0.39) (Figure 4A). Conversely, oligodendrocyte-derived EVs up-expressed COEA1 3 months POST-treatment compared with the baseline in responder patients (5770.53 ± 1003.20 and 1222.70 ± 96.04 NSAF; *p* = 0.001; FC = 4.71) (Figure 4A). These proteins are involved in the reorganisation of the extracellular matrix (ECM) (Figure 5B). Three months after treatment onset, EVs from neurons exhibited overexpression of LAMA5 in responder patients compared with PRE-treatment (7441.71 ± 963.69 and 3698.13 ± 1585.75 NSAF; *p* = 0.02; FC = 0.49) (Figure 4A), which also participates in ECM reorganisation (Figure 5B). Also, elevated levels of NonO were observed in neuron-derived EVs in responder patients compared with non-responders (6378.77 ± 1592.07 and 3094.57 ± 378.37 NSAF; *p* = 0.02; FC = 0.48) (Figure 4A), linked to an anti-inflammatory response.

Last but not least, at 3 months in responders, SPNT and NCAM were highly expressed in neuron-derived EVs compared with responders at PRE-treatment (SPNT: 2449.53 ± 687.53 and 1115.10 ± 154.66 NSAF; *p* = 0.03; FC = 0.45) (NCAM: 2040.93 ± 260.71 and 764.76 ± 191.62 NSAF; *p* = 0.002; FC = 0.37) (Figure 4A), contributing to brain repair processes such as neurite and axonal outgrowth (Figure 5B).

### 2.4. Proteins Associated with Failure of Treatment Indicate DNA Damage, Nuclear Factor Kappa B Pathway Activation, and Drug Resistance

A total of 137 proteins were identified as differentially upregulated in non-responders (Figure 5A). Notably, before treatment, PSMD7 was overexpressed in EVs secreted by B cells in non-responders compared with responders (14,666.84 ± 2773.97 and 2552.67 ± 713.15 NSAF; *p* = 0.001; FC = 5.74) (Figure 4B). Along these lines, after treatment, PRS10 was also overexpressed in EVs secreted by B cells in non-responders compared with responders (40,540.64 ± 13,602.47 and 10,425.26 ± 4146.24 NSAF; *p* = 0.02; FC = 3.88) (Figure 4B). These proteins have been shown to play a role in p53-dependent DNA damage pathways and activation of the nuclear factor kappa B (NF-κB) pathway (Figure 5B). Additionally, POST-treatment, T cell-derived EVs showed upregulation of PERM and PERL proteins in non-responders compared with responders (PERM: 35,758.18 ± 7812.72 and 17,500.57 ± 7590.68 NSAF; *p* = 0.043; FC = 2.04) (PERL: 9108.87 ± 2626.81 and 4141.23 ± 1499.29 NSAF; *p* = 0.04; FC = 2.19) (Figure 4B), which are involved in the phagocytic activity of neutrophils. Other significant proteins in Figure 4B, HS90A and RASK, were associated with drug resistance via the ERBB2 pathway.

### 2.5. Differential Protein Cargo in EVs before and after Treatment Confirms Immune Response Modulation and Cell-Matrix Adhesion Pathways in Treatment Responders

The protein cargo of PRE- and POST-treatment initiation EV samples from the responder and non-responder groups was submitted to further analysis to validate the aforementioned results. Seven enriched biological processes were identified as highly significant using FunRich software. Prior to treatment, all functions of EV proteins derived from T cells were equally regulated in both responders and non-responders. POST-treatment analysis revealed a more than twofold increase in functions related to the innate and adaptive regulation of the immune response and cell-matrix adhesion in both responders and non-responders. This enhancement, however, was not observed in EVs derived from B cells. In neuronal-derived EV proteins, functions related to immune system regulation (both innate and adaptive), cell matrix adhesion, and the regulation of the ERK1-2 cascade were equally enriched in both responders and non-responders before treatment. After treatment, these functions increased more than twofold exclusively in responders. Furthermore, in EVs derived from oligodendrocytes, these enhanced functions, as well as the negative regulation of the apoptotic process, were present only in responders in both PRE- and POST-treatment, with the exception of adaptive immune system regulation, which was represented in both groups. The differential representation of these functions in responders and non-responders before (PRE) and after (POST) treatment is shown in Figure 5C.

### 2.6. Protein–Protein Interaction Networks Predominate among Differential Proteins in Responders and Non-Responders, Highlighting Key Functional Associations

Notably, most of the differential proteins selected in responders and non-responders were found to belong to protein–protein interaction networks, which are significant in the context of functional associations. Such enrichment indicates that these proteins are at least partially biologically connected. Only NonO was found to be an unconnected protein in the responders in this analysis (Figure 6A).

### 2.7. Most Abundant Differentially Expressed Proteins in Extracellular Vesicles among Responders and Non-Responders

We also identified the most abundant differentially expressed proteins in EVs between responders and non-responders. Among the more abundant proteins, CSE1L was enriched in T cell-derived EVs from responders, whereas PSMA2 was more abundant in non-responders. Similarly, in B cell-derived EVs, PSMD6 was overexpressed in responders. EVs isolated from oligodendrocytes showed higher levels of PSB5 in responders, whereas HPRT was elevated in non-responders (Figure 6B).

## 3. Discussion

Since the early 2000s, the field of proteomics research on MS has seen explosive growth. Although many of these investigations delve into the disease pathogenesis, a significant number are dedicated to identifying molecular biomarkers across various diseases [7]. In our study, to gain insight into the differential biological functions occurring in responders and non-responders, we performed an in-depth comparative analysis of the protein content of neuron-, oligodendrocyte-, B and T cell-derived EVs. We also explored the overall statistically relevant associations with the functional processes of proteins.

The differentially overexpressed proteins identified in EVs from responders were notably associated with various immune system functions relevant to autoimmunity, including antigen processing and cross-presentation. MS is characterised by defects in tolerance mechanisms, leading to the activation of naïve autoreactive T cells through antigen cross-presentation against self-myelin proteins. These activated T cells then traverse the BBB, enter the CNS, and encounter their target antigens, thereby initiating an inflammatory response [19]. In our study, we identified 2 key proteins involved in this process: PSMD6 (which was found to be one of the most abundant proteins in EVs from B cells) and PMS5 (also highly abundant, but upregulated in EVs from oligodendrocytes), both at 3 months after treatment onset. These findings align with previous research that identified proteasome-associated proteins in EVs derived from astrocytes in animal models of experimental autoimmune encephalomyelitis (EAE) [20]. These proteins are part of the 26S proteasome subunit, a crucial component for regulated antigen degradation in the major histocompatibility complex class I. By regulating antigen presentation, these proteins likely reduce the activity of naïve autoreactive T cells targeting myelin antigens [21]. Our results suggest that in responder patients, EVs from B cells and oligodendrocytes could help mitigate the activity of naïve autoreactive T cells by diminishing the cross-presentation of myelin antigens.

Once CNS antigens are presented to autoreactive T cells, an interaction between these peripheral autoreactive T cells, along with B cells, myeloid cells, and CNS-resident cells (primarily microglia and astrocytes), induces the secretion of a range of neurotoxic inflammatory mediators, such as cytokines, chemokines, and reactive oxygen species (ROS), leading to axonal damage, neuronal demyelination, and neurovascular structure degeneration within the CNS parenchyma [22]. In our study, LV151 from the T cell-derived EVs found in responders promoted anti-inflammatory IL-4 and IL-10 cytokine synthesis, inhibiting the propagation of CNS-compartmentalised inflammatory mechanisms and, consequently, CNS injury. Taking into account that some available DMTs stimulate the production of anti-inflammatory interleukins, such as IL-4 and IL-10 [23], the EV protein content from immune system cells in patients with MS could have a role as a marker of whether DMTs are properly activating anti-inflammatory pathways, and it likely contributes to reducing CNS injury.

Within the context of CNS inflammation, we also identified other proteins that control the inflammatory response of innate immune cells in responders. We found an upregulation of NonO protein in EVs released by neurons at 3 months after treatment initiation, a transcription factor that causes inhibition of the inflammatory response in monocytes and dendritic cells [24]. In addition, NonO silences the transcription of Il-17 and Il23r by binding directly and specifically to the promoter regions of these genes, acting as a transcriptional repressor of the *Il-17* and *Il23r* genes [25]. This reinforces the role of NonO in decreasing the inflammatory response, given that IL-17 is known primarily for its ability to initiate a potent inflammatory response that includes the IL-1β, IL-6, and tumour necrosis factor [26]; and IL-23 activates macrophages and maintains chronic autoimmune inflammation via the regulation of T memory cells, especially T helper-type 17 cells, which are known to be one of the main players in orchestrating the adaptive immune response in MS [27,28]. These results could indicate that the content of NonO in neuron-derived EVs might have an anti-inflammatory effect in responders.

Our study also revealed an upregulation of the PTPRC protein in B cell-derived EVs in the responder group at 3 months POST-treatment. PTPRC plays a crucial role in regulating the overproduction of autoantibodies by B cells, a key factor in MS pathophysiology [29,30]. Previous research has linked PTPRC to MS, noting that mutations in the gene encoding this protein can predispose individuals to the disease [31]. These findings suggest that a positive treatment response could be associated with the B cell inhibition of autoreactive antibody production, addressing the aberrant immune response characteristic of MS, and that this pathway might be regulated by the content of PTTRC in B cell-derived vesicles.

Inflammation in the CNS leads to demyelination and the accumulation of myelin-toxic debris within lesions. For effective CNS repair in MS, it is essential that microglia clear this myelin debris to establish an optimal ECM and allow the initiation of brain repair processes [32]. In our study, we found that LAMA5 from neuron-derived EVs was overexpressed 3 months after treatment, and it has been shown to be linked to phagocytosis in responders through 2 mechanisms: (I) phagocytosis mediated by RAS proteins and (II) damage-associated molecular pattern-mediated TLR4 activation driven by lipopolysaccharide-binding protein (LBP). The role of both RAS family proteins and TLR4 in regulating microglial clearance of toxic myelin debris is well established [33,34,35,36]. Previous proteomics studies have highlighted the enrichment of the RAS signalling pathway in patients with MS [37] and increased TLR4 expression in EVs from patients with MS compared with healthy controls [38]. These findings suggest that EVs from responders could enhance myelin clearance through these pathways, potentially facilitating the initiation of brain repair processes.

Once the myelin molecules deposited into MS lesions are completely cleared, an optimal ECM reorganisation is needed to support axonal sprouting and recruitment of oligodendrocyte progenitor cells to undergo further differentiation to become fully functional myelin-forming oligodendrocytes that lead to remyelination in MS [32]. This structure acts as a permissive substrate, markedly enhancing synaptic plasticity, supporting neurite regrowth, axon growth, and remyelination after white matter injury [39]. Our results indicate that the EVs from oligodendrocytes and neurons containing CO6A1, COEA1, and LAMA5 in responders contribute to ECM organisation, which is likely to allow neurite and axonal sprouting and subsequent remyelination. This result is consistent with the finding that EV proteins from responders also participate in neurite outgrowth, specifically via the SPNT protein from B cell-derived EVs, a cytoplasmic adaptor protein that has a role in cell-ECM interaction, controlling ECM neural adhesion during neurite outgrowth processes [40,41]. The results are also consistent with previous findings of the proteomics analyses of other groups [42]. These results indicate that EV proteins from neurons and oligodendrocytes in patients with MS are involved in ECM remodelling and white matter repair, leading us to consider the possibility that EVs contribute to certain forms of ECM remodelling and brain plasticity in the active lesions of responders.

Our study also analysed the differential pathological functions in non-responders. As previously mentioned, the antigenic presentation of myelin proteins is a fundamental part of the pathophysiology of the disease. One of the more abundant proteins found in oligodendrocyte-derived EVs in non-responders was HPRT. Some authors have previously observed mutations in the gene encoding this protein in the T cells of patients with MS, indicating that these cells had been activated and proliferated in vivo [43]. In addition, these cell lines recognise multiple epitopes of myelin antigens [44]. Given that the oligodendrocyte is the main target cell in MS, it is likely that the HPTR protein is encapsulated by EVs during the process of antigenic recognition by T cells, reflecting a more active immunity in these patients.

In the analysis of EV protein content related to therapeutic failure, another of the characteristics found was the cell lineage involved in the phagocytotic process of myelin-toxic debris. Despite the notion that microglia are the main actors involved in this function in responders, we found that myelin debris was mainly scavenged by polymorphonuclear neutrophils in non-responders. Neutrophils are innate immune cells essential for phagocytosis; however, this process is followed by the release of large amounts of ROS [45]. When cellular production of ROS overwhelms its antioxidant capacity, it leads to a state of oxidative stress, which in turn plays a central role in several biological routes identified in our cohort of non-responders: (I) The activation of the proinflammatory NF-kB pathway by the overexpressed proteins PSMD and PRS10 in EVs from B cells, resulting in the production of NLRP3 inflammasome-derived proinflammatory IL-1, IL-6, TNF-α, and IL-12, providing a fuel that exacerbates pro-inflammatory responses in CNS lesions; (II) induction of the main pathways of cell apoptosis, including that mediated by p53, also triggered by the protein CSEL1 found in EVs from T cells and by PSMD7 and PRS10 protein release in EVs from B cells, which has been shown to be associated with the neurodegeneration that occurs in MS [46,47]; (III) regulation of ErbB2/HER2 protein expression via the HSP90 protein, classically associated with treatment resistance [48]. Moreover, in the EAE animal model, it has been shown that this biological pathway is overregulated in the worsening phase of the disease [49]. All these biological processes have been enriched in EV protein content in non-responders following phagocytosis of myelin debris by polymorphonuclear leukocytes. These results could indicate that proinflammatory processes, apoptosis, and the increase in oxidative stress are differentially activated, which can result in further CNS injury in non-responders, leading to DMT resistance in this group of patients with MS.

Lastly, we conducted a FunRich analysis to investigate how the biological processes involving proteins contained in EVs were modified before and after treatment. Our analysis revealed an enrichment of functions involved in the regulation of innate and adaptive immunity in T cell, neuronal, and oligodendrocyte EVs in responders. DMTs possess mechanisms of action that regulate different aspects of immunity, and it is known that these treatments can modulate EVs [14,15]. We hypothesise that immune system cells in responders are less resistant to DMTs, leading to the generation of a more immunosuppressive microenvironment through EVs. Additionally, we have reported that some of the most expressed proteins in responders are involved in the proper reorganisation of the ECM and inhibition of apoptosis, which aligns with the previous results of the study.

One of the primary limitations of this study is the relatively small sample size, which might limit the generalisability of the findings. Although we analysed EVs as potential biomarkers for treatment response, we did not differentiate between the effects of various treatments on the proteome of these vesicles. Different medications can have distinct influences, which could, in turn, alter EV composition and function. Our goal was to provide an initial exploration of EVs as general biomarkers of treatment response. However, future studies with larger sample sizes and a more detailed analysis of the specific effects of different treatments will be essential to validate whether the identified proteins behave consistently across diverse therapeutic approaches. Another main limitation of our study is that we employed Exoquick as the EV precipitation method, which has a risk of contamination. To mitigate this limitation, we performed a second step of immunoisolation with antibodies against EV surface markers, which rids the EV sample of contamination. Moreover, we used L1CAM as a marker of neuron-derived EVs, which has been controversial in recent years because it is also upregulated in cancer cells. Therefore, we have verified the absence of patients with cancer in our study.

## 4. Materials and Methods

### 4.1. Study Design and Participants

This single-centre, observational, longitudinal, prospective clinical study included patients diagnosed with RRMS according to the McDonald criteria [50]. Using G*Power software 3.1.9.7. (Dusseldorf, Germany), the sample size was calculated to be 80 patients with MS to achieve 80% power for No Evidence of Disease Activity-3 (NEDA-3) with a significance level of 0.05. The study was conducted from May 2021 to June 2023. The inclusion criteria were men and women older than 18 years with RRMS who were either starting a new DMT or changing their current treatment. Exclusion criteria included patients with progressive MS, current drug or alcohol dependence, severe concomitant diseases with a poor short-term prognosis, other autoimmune diseases, pregnancy or breastfeeding, and participation in pharmacological treatment trials. The study received approval from the La Paz University Hospital Research Ethics Committee (PI-4675). Data management was conducted in compliance with Spanish Law 14/2007 of July 3rd on Biomedical Research. Informed consent was obtained from all participants involved in the study.

### 4.2. Clinical Data

Participants provided demographic and clinical data, including sex, age, disease duration (time from the first MS symptom onset to the baseline visit), baseline Expanded Disability Status Scale (EDSS), and the new DMT being initiated. Standard clinical practice was followed for the required washout period for those switching from previous DMTs.

### 4.3. Outcome Measures

Each patient had 3 visits for correlative studies at specific time points: PRE-treatment (before treatment initiation) and POST-treatment (3 and 12 months after treatment onset). Figure 1 shows the variables collected at each visit. Clinical assessments are those agreed upon by MS neurologists for monitoring DMT response [51]:Relapses: New or recurrent neurological symptoms not associated with fever, lasting for at least 24 h, followed by 30 days of stability or improvement.MRI Activity: Presence of one or more new or enlarged lesions on a T2-weighted scan at 12 months.Disease Progression: An increase of 1.5 points in the EDSS score if the baseline EDSS score was 0; an increase of 1.0 point for baseline EDSS scores between 1 and 5.5; an increase of 0.5 points from baseline EDSS ≥6; or a 20% increase in the 9-Hole Peg Test (9-HPT) at 12 months compared to baseline.

All clinical assessments and outcome measurements were conducted by experienced neurologists.

### 4.4. Treatment Response

Treatment response was assessed at the 12-month follow-up according to the European Committee for Treatment and Research in Multiple Sclerosis (ECTRIMS) guidelines and the NEDA-3 criteria [3]. NEDA-3 is defined as the absence of relapses, no MRI activity, and no disability progression over the last 12 months. In clinical practice, as well as in this study, NEDA-3 differentiates between patients who respond to DMT (“responders”) and those who do not (“non-responders”).

### 4.5. Extracellular Vesicle Isolation

Blood samples were collected at PRE-treatment and at 3 months POST-treatment in 9 mL EDTA tubes. Samples were centrifuged at 1600 g for 15 min to obtain cell-free plasma, which was then stored at −80 °C until analysis. For EV isolation, 1 mL of plasma per sample was thawed rapidly in a 37 °C water bath, transferred to a 1.5 mL Eppendorf tube, and centrifuged at 2000 g for 20 min at room temperature. EV purification targeted those derived from neurons, oligodendrocytes, B cells, and T cells, using a 2-step isolation procedure involving precipitation and immune isolation. We used the ExoQuick EV isolation kit (System Biosciences, Palo Alto, CA, USA) for precipitation, followed by immune-isolation with biotinylated antibodies against specific EV surface markers. Antibodies included anti-L1CAM for neuronal EVs (Thermo Fisher Scientific, Waltham, MA, USA), anti-MOG for oligodendrocyte EVs (R&D Systems, Minneapolis, MI, USA), anti-CD20 for B cell EVs (Merck Millipore, Hesse, Germany), and anti-CD3 for T cell EVs (Merck Millipore, Hesse, Germany), as previously described [16]. Streptavidin agarose beads (Thermo Fisher Scientific, Waltham, MA, USA) were used to precipitate EVs, which were then stored at −80 °C until further use.

### 4.6. Extracellular Vesicle Characterisation

For characterisation, EVs from neurons, oligodendrocytes, B cells, and T cells were thawed on ice and analysed for the following:Specific EV markers: Western blot analysis was performed using antibodies against CD9 (Thermo Fisher Scientific, Waltham, MA, USA), CD81 (1:250, Abcam, Cambridge, UK), and CD63 (1:250, Abcam, Cambridge, UK), followed by goat anti-mouse or anti-rabbit HRP antibodies (1:750, Invitrogen, Waltham, MA, USA). Blots were visualised using Pierce ECL chemiluminescence (Thermo Fisher Scientific, Waltham, MA, USA) and a UVITEC–Cambridge imaging system, as previously described [16].EV morphology: Transmission electron microscopy (JEOL JEM1010) was used to analyse EV morphology as previously described [16].EV size and concentration: NTA was performed using the NanoSight NS500 nanoparticle analyser (Malvern Instruments, Worcestershire, UK), equipped with fast video capture and particle-tracking software. Measurements were conducted in triplicate, following the manufacturer’s instructions, as previously described [16].

### 4.7. Proteomics Analysis

We performed an in-depth analysis of the protein content of neuron-, oligodendrocyte-, B, and T cell-derived EVs, comparing responders and non-responders based on NEDA-3 parameters to gain insight into the differential biological functions that occur across the study groups. To build on this, we prepared 3 sample pools as biological replicates from both groups to obtain a relevant biological average.

#### 4.7.1. Protein Tryptic Digestion

For trypsin digestion, an equal amount of protein from all samples was loaded on a 10% SDS-PAGE gel. The run was interrupted when the front penetrated 3mm into the resolving gel [16,52]. The protein band was visualised with Sypro-Ruby fluorescent staining (Lonza, Pontevedra, Spain), excised, and processed for in-gel tryptic digestion following the standard protocol [53]. In addition, 10 mM dithiothreitol (Sigma-Aldrich, St. Louis, MO, USA) in 50 mM ammonium bicarbonate (Sigma-Aldrich, St. Louis, MO, USA) was used for gel pieces reduction and 55 mM iodoacetamide (Sigma-Aldrich, St. Louis, MO, USA) in 50 mM ammonium bicarbonate for alkylation. Next, pieces were washed with 50 mM ammonium bicarbonate in 50% methanol (HPLC grade, Barcelona, Spain), dehydrated by adding acetonitrile (HPLC grade, Barcelona, Spain), and dried in a SpeedVac. At last, modified porcine trypsin (Promega, Madison, WI, USA) was added at a concentration of 20 ng/μL in 20 mM ammonium bicarbonate and incubated overnight at 37 °C. Peptides were extracted by three 20 min incubations in 40 μL of 60% acetonitrile in 0.5% HCOOH and stored at −20 °C.

#### 4.7.2. Mass Spectrometric Analysis

Data-Dependent Acquisition (DDA). Digested peptides were resuspended in mobile phase A (0.1%, formic acid in water) by sonication for 10 min to obtain 1 μg/μL peptide solution. The gradient was created using a micro-liquid chromatography system (Eksigent Technologies nanoLC 400, Sciex, Framingham, MA, USA) coupled to a high-speed Triple TOF 6600 mass spectrometer (Sciex, Framingham, MA, USA) with a microflow source. Peptides (4 µg of each sample) were separated using reverse-phase chromatography, using a ChromXP C18CL analytical column (150 μL ×0.3mm, 120Å, s-3 μL) (Sciex, Framingham, MA, USA). The trap column was a YMC-TRIART C18 (YMC Technologies, Teknokroma) with a 3 mm particle size and 120 Å pore size, switched online with the analytical column. The loading pump delivered a solution of 0.1% formic acid in water at 10 µL/min. The micro-pump generated a flow rate of 5 µL/min and was operated under gradient elution conditions. For that, 0.1% formic acid in water was used as mobile phase A and 0.1% formic acid in acetonitrile as mobile phase B. Peptides were separated using a 90 min gradient ranging from 2% to 90% mobile phase B. When the peptides eluted, they were directly ionised and injected into a hybrid quadrupole-TOF mass spectrometer Triple TOF 6600 (Sciex, Framingham, MA, USA) operated with a DDA system in positive ion mode. A Micro source (Sciex, Framingham, MA, USA) was used for the interface between micro-LC and MS, with an application of 2600 V voltage. The acquisition mode consisted of a 250 ms survey MS scan from 400 to 1250 *m*/*z* followed by an MS/MS scan from 100 to 1500 *m*/*z* (25 ms acquisition time) of the top 65 precursor ions from the survey scan, for a total cycle time of 2.8 s. The fragmented precursors were then added to a dynamic exclusion list for 15 s; any singly charged ions were excluded from the MS/MS analysis. The instrument was automatically calibrated every 4 h using external calibrant tryptic peptides from PepCalMix solution (AB Sciex, Framingham, MA, USA) [16,52].

#### 4.7.3. Protein Quantification by SWATH-MS (Sequential Window Acquisition for All Theoretical Mass Spectra)

Spectral library creation. To build the MS/MS spectral libraries, peptide solutions were analysed by a DDA method using micro-LC-MS/MS as described before. Three sample pools as biological replicates from both groups were made using 3 μL of each sample to obtain a general representation of the peptides and proteins present in all samples. Then, 4 μL of each pool was separated into a micro-LC system Ekspert nLC425 (Eksigen, Temecula, CA, USA) as described before but using a 40 min gradient. All the mass spectrometry parameters were those used in the DDA analysis described before. After the MS/MS analyses, data files processing was performed using Protein Pilot software (version 5.0.2, Sciex, Redwood City, CA, USA) searched against a Human specific Uniprot database (accessed on 2 February 2023) (https://www.uniprot.org/) (UniProt release 2022_02 With 44413 human proteins), specifying iodoacetamide as Cys alkylation and Trypsin as an enzyme used in digestion. The software uses the algorithm ParagonTM for database search and ProgroupTM (accessed on 27 January 2023) for data grouping [16,52]. The false discovery rate (FDR) was set to 1% for both peptides and proteins, using a non-lineal fitting method. The MS/MS spectra of the identified peptides were then used to generate the spectral library for SWATH peak extraction using the plug-in MS/MSALL with SWATH Acquisition MicroApp (version 2.0, Sciex) for PeakView Software (version 2.2, Sciex, Redwood City, CA, USA). Peptides with a confidence score above 99% (as obtained from Protein Pilot database search) (accessed on 27 January 2023) were included in the spectral library.

Relative quantification by SWATH-MS acquisition. Regarding relative quantification by SWATH-MS acquisition, 4 μL of each sample was analysed in a data-independent acquisition method (DIA). The method is based on repeating a cycle consisting of the acquisition of 100 sequential overlapping precursor isolation windows of variable width (1 *m*/*z* overlap), covering the mass range from 400 to 1250 *m*/*z* with a previous TOF MS1 scan (400 to 1500 *m*/*z*, 50 ms acquisition time) for each cycle. The total cycle time was 6.3 s.

#### 4.7.4. Data Analysis

Data extraction of chromatographic fragment ion profiles from the SWATH method was performed with PeakView software (version 2.2; Sciex, Redwood City, CA, USA) using the SWATH AcquisitionMicroApp (version 2.0; Sciex, Redwood City, CA, USA), which processed the data using the spectral library created from the DDA data. Five-minute windows and 30 ppm widths were used to extract the ion chromatograms; SWATH quantization was attempted for all proteins in the ion library that were identified by ProteinPilot with an FDR below 1%. The retention times from the peptides that were selected for each protein were realigned in each run according to the iRT peptides present in the samples and eluted along the whole-time axis. The extracted ion chromatograms were then generated for each selected fragment ion; the peak areas for the protein were obtained by summing the peak areas from 10 peptides (MS1 scan) and 7 corresponding fragment ions (MS2 scan) from each peptide. PeakView computed an FDR and a score for each assigned peptide according to the chromatographic and spectra components; only peptides with an FDR below 1 % were used for protein quantization. Protein quantization was calculated by adding the peak areas of the corresponding peptides. Integrated peak areas were exported directly to MarkerView software 1.3.1. (Sciex, Framingham, MA, USA) for relative quantitative analysis. The export generated three files containing quantitative information about individual ions, the summed intensity of different ions for a particular peptide, and the summed intensity of different peptides for a particular protein. MarkerView uses processing algorithms that accurately find chromatographic and spectral peaks directly from raw SWATH data. Data alignment via MarkerView compensates for small variations in mass and retention time values, ensuring that identical compounds from different samples are accurately compared to each other. A most-like ratio (MLR) normalization was performed after statistical analysis to control for possible uneven sample loss across the different samples during the sample preparation process [16,52].

### 4.8. Biological Functions and Pathway Study

To elucidate possible mechanisms governing treatment response, we explored the potential biological functions in which the proteins identified in responders and non-responders were involved. To this end, we used the Reactome pathway database (accessed on 16 August 2023) (https://reactome.org) for visualising, interpreting, and analysing the differential biological processes in responders and non-responders in which differentially identified proteins are involved [54].

### 4.9. Functional Enrichment Analysis

We explored the functional enrichment analysis of the proteins identified using FunRich software 3.1.4. to evaluate statistically relevant associations with the functional processes of the selected proteins [55].

### 4.10. Protein–Protein Interaction Network Analysis

On the basis of the proteomic analysis, we then assessed protein–protein interactions including physical and functional associations in responders versus non-responders using the STRING database (accessed on 24 October 2023) (http://string-db.org). The interactions in STRING originate in 5 main sources: genomic context predictions, high-throughput experiments, co-expression, and automated text mining [56].

### 4.11. Statistics

The statistical analysis was performed using SPSS 23.0 for Windows (SPSS Inc., IBM, Chicago, IL, USA). Categorical variables were described as percentages, and proportions between groups were compared using the chi-squared test; Fisher’s exact test was used for dichotomous variables. Continuous variables were expressed as mean (standard deviation) or median (interquartile range). A t-test and analysis of variance with Bonferroni post hoc correction were used for multiple comparisons of normally distributed data. Kruskal–Wallis or Mann–Whitney U tests were used for the comparison of non-normally distributed data sets. Multivariate statistical analysis through principal component analysis (PCA) was performed to compare sample data. The mean MS peak area of each protein was obtained from the SWATH-MS replicates of each condition. The cut-off for the statistically significant protein change was a *p*-value ≤ 0.05 and a fold change of >2. The data were represented using GraphPad Prism 8 software (GraphPad software, La Jolla, CA, USA) and Adobe Illustrator (Adobe Inc., San Jose, CA, USA, USA).

## 5. Conclusions

This study introduces a novel EV protein profiling approach as a biomarker for predicting treatment response in MS. Treatment efficacy can be predicted if the patient’s T cell-derived EVs contain LV151 prior to treatment, their B cell-derived EVs contain PMSD6 and PTPRC upregulated, oligodendrocyte-derived EVs have CO6A1 and COEA1, and neuron-derived EVs exhibit LAMA5, NonO, SPNT, and NCAM, 3 months after treatment initiation. Lack of response to treatment can be anticipated if the patient B cell-derived EVs contain PSMD7 and PRS10 overexpressed and if their T cell-derived EVs have PERM and PERL at 3 months. This article also provides relevant insights into the pathogenesis that influences patient response to treatment. Key findings include the role of EVs in modulating immune responses and promoting CNS repair in responders, particularly through the regulation of antigen presentation, anti-inflammatory cytokine synthesis, and ECM reorganisation. In contrast, non-responders displayed an upregulation of pathways associated with inflammation, apoptosis, and treatment resistance. This strategy facilitates the recognition of treatment failure early in the treatment initiation phase, as opposed to a blind and prolonged follow-up process characterised by waiting for a clinical relapse, disease progression, or the emergence of new abnormal MRI findings. This marks a step towards personalised medicine for this immune-mediated, demyelinating, and inflammatory disease.

## Figures and Tables

**Figure 1 ijms-25-10761-f001:**
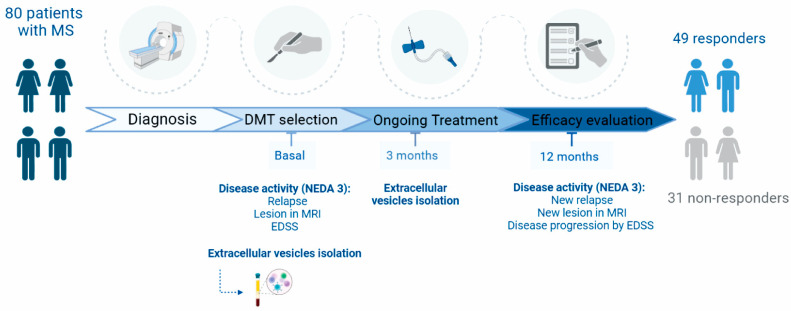
Flow diagram of the study illustrating the procedures applied to patients with MS and their distributions into groups. The blue figures represent responders, while the gray figure indicates non-responders. Abbreviations: EDSS, expanded disability status scale; MRI, magnetic resonance imaging; NEDA, no evidence of disease activity; EV, extracellular vesicles; MS, multiple sclerosis.

**Figure 2 ijms-25-10761-f002:**
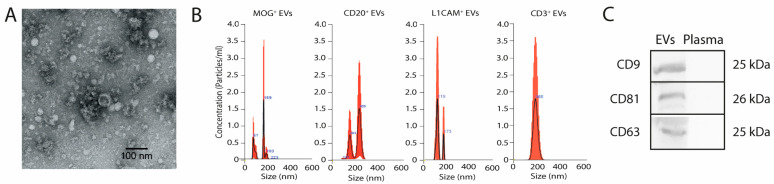
(**A**) Electron microscope image of EVs smaller than 200 nm. (**B**) Size dispersion and concentration of EV samples analysed by NTA. (**C**) Western blot image demonstrating the positivity of specific markers CD9, CD81, and CD63 in the EV membrane. Negative control samples are from plasma. The gel image was cropped. Abbreviations: EV, extracellular vesicles; kDa, Kilodalton.

**Figure 3 ijms-25-10761-f003:**
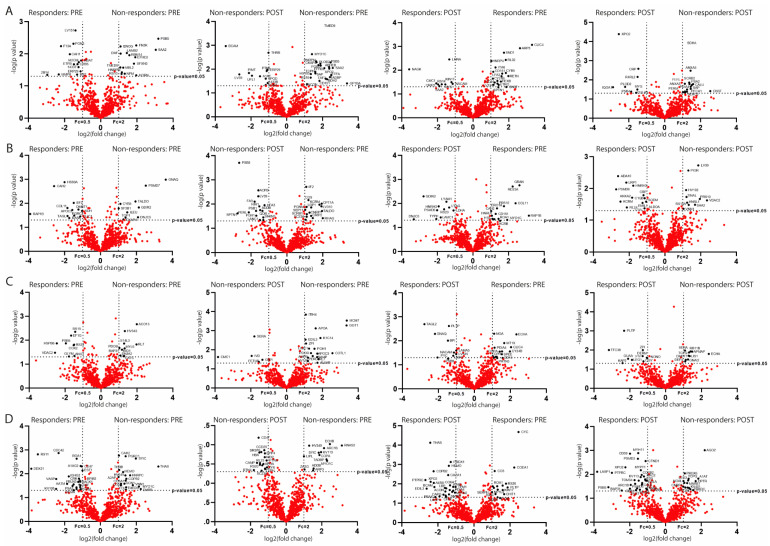
The differentially expressed proteins analysed by volcano plots between responders and non-responders in PRE and POST in (**A**) T cells, (**B**) B cells, (**C**) neurons, and (**D**) oligodendrocytes. The red dots represent proteins common to both groups, while the black dots represent differentially expressed proteins.

**Figure 4 ijms-25-10761-f004:**
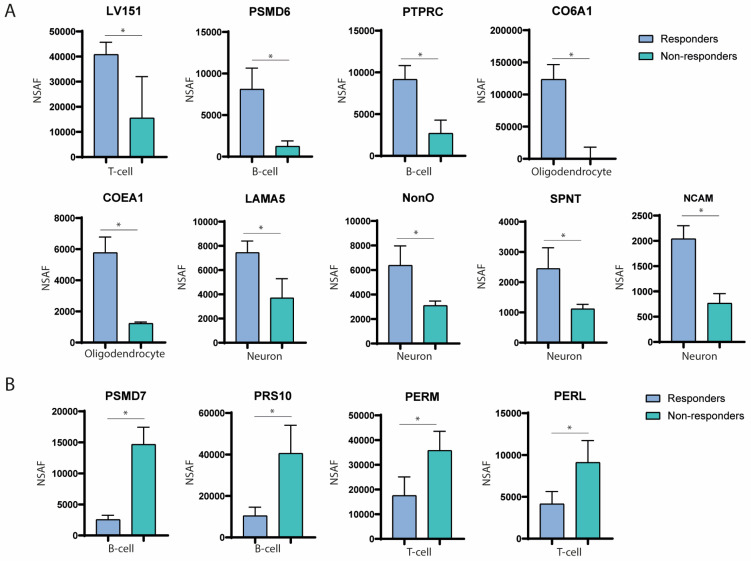
(**A**) Differentially upregulated proteins identified in responders. (**B**) Most differentially overexpressed proteins in non-responders. The blue color represents responder patients, while the green color indicates non-responder patients. * *p* < 0.05.

**Figure 5 ijms-25-10761-f005:**
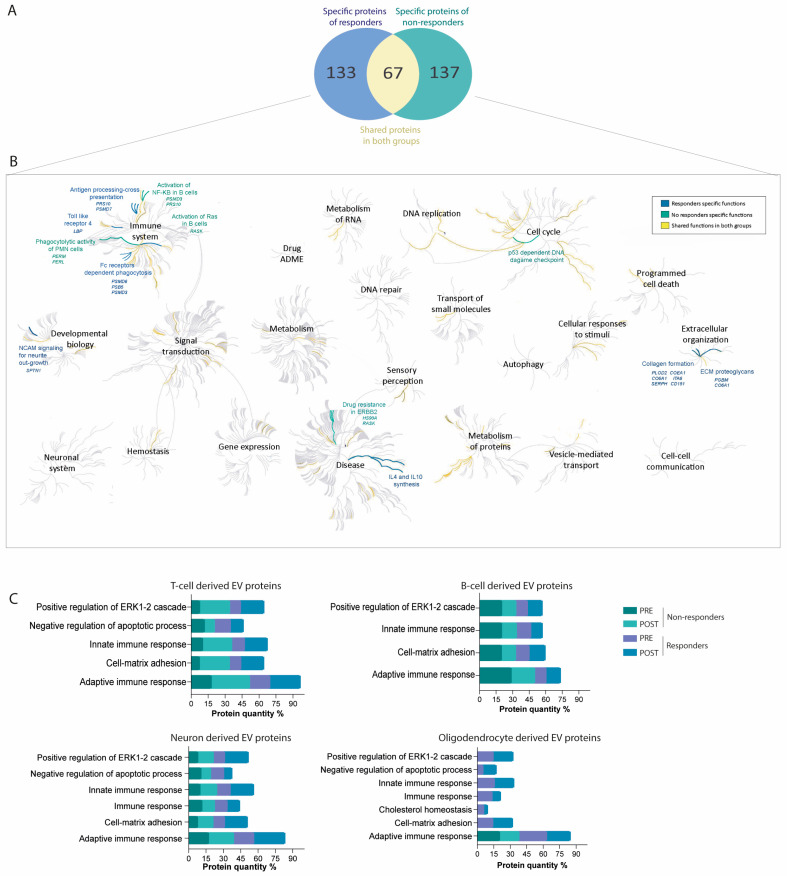
(**A**) Venn diagram showing the differentially expressed and shared proteins in responders and non-responders. (**B**) Pathway enrichment analysis using reactome databases (accessed on 16 August 2023) of 133 differentially unregulated proteins from responders (in blue) and 137 proteins from non-responders (in green). (**C**) Functional enrichment analysis using FunRich tool of the upregulated proteins in T cells, B cells, neurons, and oligodendrocytes in EVs in PRE- and POST-treatment of responders and non-responders. The blue color represents functions present in responder patients, the green color indicates functions in non-responder patients, and the yellow color represents those common to both groups.

**Figure 6 ijms-25-10761-f006:**
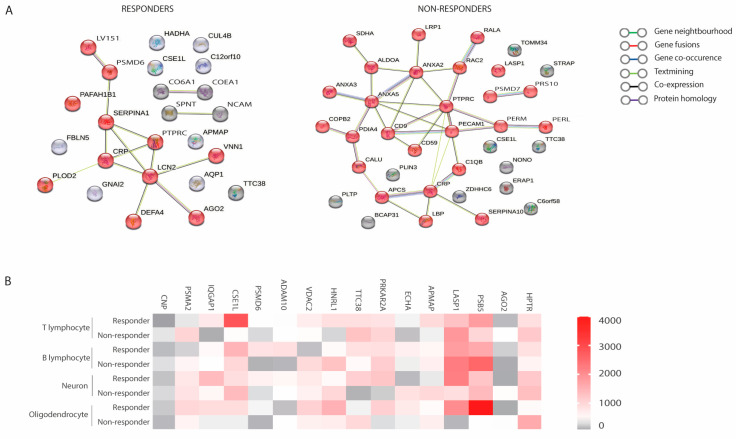
(**A**) Protein–protein interaction network for proteins described in responders and non-responders. (**B**) Heatmap analysis of differentially expressed proteins in responders vs. non-responders in T cells, B cells, neurons, and oligodendrocytes. In the colour bar, red represents high expression, and grey represents low expression.

**Table 1 ijms-25-10761-t001:** Demographic and clinical data of the study participants.

	Responders (*n* = 49)	Non-Responders (*n* = 31)	*p*-Value
**Demographics**			
Women, *n* (%)	25 (51%)	18 (58.1%)	0.34
Age, years	44.06 (9.03)	43.82 (11.23)	0.91
**Clinical data**			
Time from diagnosis, months	123.53 (120.48)	153.3 (137.01)	0.30
Baseline EDSS	1.89 (2.07)	2 (2.29)	0.88
**Treatments received**			0.59
Natalizumab, *n* (%)	12 (24.5)	6 (19.4)	
Teriflunomide, *n* (%)	2 (4.1)	3 (9.7)	
Interferon, *n* (%)	6 (12.2)	0 (0)	
Dimethyl fumarate, *n* (%)	7 (14.3)	6 (19.4)	
Ocrelizumab, *n* (%)	7 (14.2)	3 (9.7)	
Siponimod, *n* (%)	1 (2.0)	0 (0)	
Cladribine, *n* (%)	14 (28.6)	11 (35.5)	

All values are mean (SD) unless otherwise noted; Mann–Whitney U test for continuous variables and Fisher’s exact test for categorical variables were employed to determine statistically significant differences between groups. Abbreviations: *n*, number; SD, standard deviation; EDSS, expanded disability status scale.

## Data Availability

The mass spectrometry proteomics data have been deposited to the ProteomeXchange Consortium via the PRIDE partner repository with dataset identifier PXD048002.

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
