# Peer review of "Differential Protein Expression in Extracellular Vesicles Defines Treatment Responders and Non-Responders in Multiple Sclerosis"

_ijms, 2024, doi:10.3390/ijms251910761_

Round 1

Reviewer 1 Report

Comments and Suggestions for Authors

Your research on the role of extracellular vesicles (EVs) in multiple sclerosis is very promising and the data presented are robust.

1. In the abstract, it is essential to indicate whether CO6A1 and COEA1 proteins are increased in patients who respond to treatment and whether PERM and PERL proteins are elevated in patients who do not respond to treatment. In addition, include a description of NF-κB (Nuclear Factor kappa B).

2. Include the ethics committee data in the methods section to ensure ethical compliance of the research.

3. In item 4.5, both rpm and g are used to describe centrifugation. I suggest standardizing the unit of measurement for gravitational force (g), since the rotation per minute (rpm) depends on the radius of the centrifuge. The same rpm can result in different gravitational forces (g) depending on the radius of the rotor.

4. The electron microscope image B), which illustrates an EV in Figure 1, is of poor quality. If possible, replace it with an image of higher resolution and size to allow a clearer visualization of the structure and scale.

Author Response

Reviewers’ Comments to the manuscript entitled "Differential Protein Expression in Extracellular Vesicles Defines Treatment Responders and Non-Responders in Multiple Sclerosis" by Gabriel Torres Iglesias, MariPaz López Molina, Lucía Botella, Fernando Laso-García, Beatriz Chamorro, Mireya Fernández-Fournier, Inmaculada Puertas, Susana B. Bravo, Elisa Alonso López, Exuperio Diez-Tejedor, María Gutiérrez-Fernández, Laura Otero-Ortega.

===============================================================

Reviewer #1:

Your research on the role of extracellular vesicles (EVs) in multiple sclerosis is very promising and the data presented are robust.

Thank you for your comments and suggestions. They will certainly help to improve our manuscript.

  1. In the abstract, it is essential to indicate whether CO6A1 and COEA1 proteins are increased in patients who respond to treatment and whether PERM and PERL proteins are elevated in patients who do not respond to treatment. In addition, include a description of NF-κB (Nuclear Factor kappa B).

Thank you for your valuable suggestion. We have addressed your comment by including the requested information in lines 28, 31, and 32 of the abstract. Specifically, we now indicate whether CO6A1 and COEA1 proteins are increased in patients who respond to treatment, and whether PERM and PERL proteins are elevated in patients who do not respond to treatment. Additionally, a description of NF-κB (Nuclear Factor kappa B) has been incorporated into the abstract.

  1. Include the ethics committee data in the methods section to ensure ethical compliance of the research.

Thank you for your observation. We have included the ethics committee data in the Materials and Methods section, specifically under the Study Design and Participants subheading, in lines 423 and 426, to ensure the ethical compliance of the research.

  1. In item 4.5, both rpm and g are used to describe centrifugation. I suggest standardizing the unit of measurement for gravitational force (g), since the rotation per minute (rpm) depends on the radius of the centrifuge. The same rpm can result in different gravitational forces (g) depending on the radius of the rotor.

We have standardized the unit of measurement for gravitational force by replacing "rpm" with "g" in line 464 to avoid any confusion, as the gravitational force can vary depending on the radius of the centrifuge.

  1. The electron microscope image B), which illustrates an EV in Figure 1, is of poor quality. If possible, replace it with an image of higher resolution and size to allow a clearer visualization of the structure and scale.

We have changed the electron microscope image in the new Figure 2, in order to clear visualization of structure.

We appreciate your careful review and believe these updates improve the clarity of our findings.

Reviewer 2 Report

Comments and Suggestions for Authors

The authors presented a Label-free quantitative proteomic profiling approach for characterization the significance levels changes proteins in Extracellular Vesicles between Responders and Non-Responders treatment in patients with Multiple Sclerosis. The presented method and the analysis of the acquired results are in compliance with proteomics approaches guideline.

However, several details in the Methods and results section are either missing, misrepresented, disorganized or have errors. I recommend the authors proofread the whole article to make sure the experimental procedures and results are clearly presented. Here are some examples of misrepresented information in this manuscript:

Main concern:

1.            The patients treated by different medicine as mention in table 1 and as we know each medicine reflect on different way in the metabolism of human body, that affect the proteome and, on the results, and also in the response of the treatment. Please clarify.

2.            Why the authors decide (reason) to stop the treatment in the 3 month and no more or less

3.            Authors doesn’t mention the inclusion and exclusion criteria of patient’s recruitment.

4.            Line 69 and 74 provide the references for the genomics and proteomics previous study.

5.            In figure 1 should be separated the flow diagram alone and Characterization of brain- and immune system-derived EVs result should be presented with high resolution in one figure.

6.            Figure 2 its confused need to be described well in the results section and provide a table content all information regarding the proteins significantly changed between all the group in supplementary section. 

7.            In proteomics analysis section missing: Sample preparation for proteomics, Liquid chromatography coupled with tandem mass spectrometry (LC‒MS/MS), Data processing…etc.

8.            Authors should mention the cut off of the statistically significant change of the proteins used as p-value and fold change.

9.            FDRs used for the protein and peptide are missing.

10.          The article needs to be reviewed for syntax, punctuation and Grammer.

Comments on the Quality of English Language

The article needs to be reviewed for syntax, punctuation and Grammer.

Author Response

Reviewers’ Comments to the manuscript entitled "Differential Protein Expression in Extracellular Vesicles Defines Treatment Responders and Non-Responders in Multiple Sclerosis" by Gabriel Torres Iglesias, MariPaz López Molina, Lucía Botella, Fernando Laso-García, Beatriz Chamorro, Mireya Fernández-Fournier, Inmaculada Puertas, Susana B. Bravo, Elisa Alonso López, Exuperio Diez-Tejedor, María Gutiérrez-Fernández, Laura Otero-Ortega.

=============================================================================

Reviewer #2:

The authors presented a Label-free quantitative proteomic profiling approach for characterization the significance levels changes proteins in Extracellular Vesicles between Responders and Non-Responders treatment in patients with Multiple Sclerosis. The presented method and the analysis of the acquired results are in compliance with proteomics approaches guideline.

Thank you for your comments and suggestions. They will certainly help to improve our manuscript.

However, several details in the Methods and results section are either missing, misrepresented, disorganized or have errors. I recommend the authors proofread the whole article to make sure the experimental procedures and results are clearly presented. Here are some examples of misrepresented information in this manuscript:

 Main concern:

  1. The patients treated by different medicine as mention in table 1 and as we know each medicine reflect on different way in the metabolism of human body, that affect the proteome and, on the results, and also in the response of the treatment. Please clarify.

We greatly appreciate your insightful comments. Indeed, as you have pointed out, different medications can influence human metabolism in various ways, potentially affecting the proteome and, consequently, the results and treatment response. In our study, we have specifically analyzed EVs as biomarkers, which may indeed be modulated by the type of treatment administered. However, our primary aim was to take an initial step in investigating EVs as a general biomarker for treatment response, rather than focusing on the effects of different treatments individually.

We fully agree that, with a larger sample size, it would be highly valuable to explore whether these proteins behave similarly across different treatments, and this will certainly be an important aspect for future research. We have now included this consideration in the limitations section of the study (lines 397-405). Now it reads as follows:

“One of the primary limitations of this study is the relatively small sample size, which might limit the generalisability of the findings. Although we analysed EVs as potential biomarkers for treatment response, we did not differentiate between the effects of various treatments on the proteome of these vesicles. Different medications can distinctly influence, which could, in turn, alter EV composition and function. Our goal was to provide an initial exploration of EVs as general biomarkers of treatment response. However, future studies with larger sample sizes and a more detailed analysis of the specific effects of different treatments will be essential to validate whether the identified proteins behave consistently across diverse therapeutic approaches.”

  1. Why the authors decide (reason) to stop the treatment in the 3 month and no more or less

Thank you for your question, and we apologize for any misunderstanding. The treatment is not stopped at the 3-month mark. Instead, a blood sample is collected at this time point because it is when the treatment begins to take effect. The response to treatment is evaluated over the course of a full year. While the study spans one year, patients continue their treatment as per routine clinical practice, without any interruptions for the study. The decision to collect the blood sample at 3 months aims to identify early biomarkers of therapeutic response, allowing us to assess the effectiveness of the treatment before a full year has passed and before any potential treatment failure could lead to clinical relapse, new MRI lesions, or increased disability. This information is reflected in 644 to 647 lines of the manuscript.

  1. Authors doesn’t mention the inclusion and exclusion criteria of patient’s recruitment.

The inclusion and exclusion criteria for patient recruitment were outlined in lines 423 to 426 of the manuscript. However, if you prefer a more detailed explanation, we would be happy to include more information in the Materials and Methods section for further clarity.

  1. Line 69 and 74 provide the references for the genomics and proteomics previous study.

We have included the following references:

- Sandi, D.; Kokas, Z.; Biernacki, T.; Bencsik, K.; Klivényi, P.; Vécsei, L. Proteomics in Multiple Sclerosis: The Perspective of the Clinician. Int. J. Mol. Sci. 2022, 23, 5162. https://doi.org/10.3390/ijms23095162.

- Drabik, A; Bierczynska-Krzysik, A; Bodzon-Kulakowska, A; Suder, P; Kotlinska, J; Silberring, J. Proteomics in neurosciences. Mass Spectrom Rev. 2007, 26:432-50. https://doi: 10.1002/mas.20131.

  1. In figure 1 should be separated the flow diagram alone and Characterization of brain- and immune system-derived EVs result should be presented with high resolution in one figure.

Thank you for your comment. We have addressed your suggestion by dividing Figure 1 into two separate figures:

Figure 1: Flow chart.

Figure 2: EV Characterization, presented with high resolution.

We believe this improves clarity and presentation of the data. 

  1. Figure 2 its confused need to be described well in the results section and provide a table content all information regarding the proteins significantly changed between all the group in supplementary section. 

Thank you for your suggestion. In order to clarify, we have prepared a table that includes all the differentially expressed proteins between responders and non-responders, at both PRE and POST treatment time points, and from different cell origins (T cells, B cells, oligodendrocytes, and neurons). This table has been added to the supplementary section. If you need further clarification, we are happy to make any additional information as needed.

  1. In proteomics analysis section missing: Sample preparation for proteomics, Liquid chromatography coupled with tandem mass spectrometry (LC‒MS/MS), Data processing…etc.

Thank you for your observation, and we apologize for the oversight. We have now included more detailed information on the proteomics analysis section in 502-595 lines. Now it reads as follows:

4.7.1. Protein tryptic digestion

For trypsin digestion, an equal amount of protein from all samples was loaded on a 10% SDS-PAGE gel. The run was interrupted when the front penetrated 3mm into the re-solving gel [16,52] The protein band was visualised, with Sypro-Ruby fluorescent staining (Lonza, Spain), excised, and processed for in-gel tryptic digestion following the standard protocol [53]. 10 mM dithiothreitol (Sigma-Aldrich, USA) in 50 mM ammonium bicar-bonate (Sigma-Aldrich, USA) was used for gel pieces reduction and 55 mM iodoacetamide (Sigma-Aldrich, USA) in 50 mM ammonium bicarbonate for alkylation. Next, pieces were washed with 50 mM ammonium bicarbonate in 50% methanol (HPLC grade, Spain), de-hydrated by adding acetonitrile (HPLC grade, Spain), and dried in a SpeedVac. At last, modified porcine trypsin (Promega, USA) was added at a concentration of 20 ng/μL in 20 mM ammonium bicarbonate and incubated overnight at 37 ºC. Peptides were extracted by three 20 min incubations in 40 μL of 60% acetonitrile in 0.5% HCOOH and stored at -20°C.

4.7.2. Mass spectrometric analysis

Data-dependent Acquisition (DDA). Digested peptides were resuspended in mobile phase A (0.1%, formic acid in water), by sonication for 10 min to obtain 1μg/μL peptide solution. The gradient was created using a micro-liquid chromatography system (Eksigent Technologies nanoLC 400, Sciex, USA) coupled to a high-speed Triple TOF 6600 mass spectrometer (Sciex, USA) with a microflow source. Peptides (4 µg of each sample) were separated using reverse-phase chromatography, using a ChromXP C18CL analytical column (150 μL ×0.3mm, 120Å, s-3 μL) (Sciex, USA). The trap column was a YMC-TRIART C18 (YMC Technologies, Teknokroma) with a 3 mm particle size and 120 Å pore size, switched online with the analytical column. The loading pump delivered a so-lution of 0.1% formic acid in water at 10 µl/min. The micro-pump generated a flow rate of 5 µl/min and was operated under gradient elution conditions. For that, 0.1% formic acid in water was used as mobile phase A, and 0.1% formic acid in acetonitrile as mobile phase B. Peptides were separated using a 90 min gradient ranging from 2% to 90% mobile phase B. When the peptides eluted, they were directly ionized and injected into a hybrid quadrupole-TOF mass spectrometer Triple TOF 6600 (Sciex, USA) operated with a DDA system in positive ion mode. A Micro source (Sciex, USA) was used for the interface be-tween micro-LC and MS, with an application of 2600 V voltage. The acquisition mode consisted of a 250 ms survey MS scan from 400 to 1250 m/z followed by an MS/MS scan from 100 to 1500 m/z (25 ms acquisition time) of the top 65 precursor ions from the survey scan, for a total cycle time of 2.8 seconds. The fragmented precursors were then added to a dynamic exclusion list for 15 seconds; any singly charged ions were excluded from the MS/MS analysis. The instrument was automatically calibrated every 4 hours using exter-nal calibrant tryptic peptides from PepCalMix solution (AB Sciex, USA) [16,52].

4.7.3. Protein quantification by SWATH-MS (Sequential Window Acquisition for All Theoretical Mass Spectra).

Spectral library creation. To build the MS/MS spectral libraries, peptide solutions were analyzed by a DDA method using micro-LC-MS/MS as described before. Three sam-ple pools as biological replicates from both groups were made using 3 μL of each sample to obtain a general representation of the peptides and proteins present in all samples. 4 μL of each pool was separated into a micro-LC system Ekspert nLC425 (Eksigen, USA) as de-scribed before but using a 40 minutes gradient. All the mass spectrometry parameters were those used in the DDA analysis described before. After MS/MS analyses, data files processing was performed using Protein Pilot software (version 5.0.2, Sciex, USA) searched against a Human specific Uniprot database (https://www.uniprot.org/) (UniProt release 2022_02 With 44413 human proteins), specifying iodoacetamide as Cys alkylation and Trypsin as enzyme used in digestion. The software uses the algorithm ParagonTM for database search and ProgroupTM for data grouping [16,52]. The false discovery rate (FDR) was set to 1% for both peptides and proteins, using a non-lineal fitting method. The MS/MS spectra of the identified peptides were then used to generate the spectral library for SWATH peak extraction using the plug-in MS/MSALL with SWATH Acquisition Micro-App (version 2.0, Sciex) for PeakView Software (version 2.2, Sciex, USA). Peptides with a confidence score above 99% (as obtained from Protein Pilot database search) were includ-ed in the spectral library.

Relative quantification by SWATH-MS acquisition. Regarding relative quantification by SWATH-MS acquisition, 4μL of each sample was analyzed in a data-independent ac-quisition method (DIA). The method is based on repeating a cycle consisting of the acqui-sition of 100 sequential overlapping precursor isolation windows of variable width (1 m/z overlap), covering the mass range from 400 to 1250 m/z with a previous TOF MS1 scan (400 to 1500 m/z, 50 ms acquisition time) for each cycle. The total cycle time was 6.3 s.

4.7.4. Data analysis

Data extraction of chromatographic fragment ion profiles from the SWATH method was performed with PeakView software (version 2.2; Sciex, USA) using the SWATH Ac-quisitionMicroApp (version 2.0; Sciex, USA), which processed the data using the spectral library created from the DDA data. Five-minute windows and 30 ppm widths were used to extract the ion chromatograms; SWATH quantization was attempted for all proteins in the ion library that were identified by ProteinPilot with an FDR below 1%. The retention times from the peptides that were selected for each protein were realigned in each run ac-cording to the iRT peptides present in the samples and eluted along the whole-time axis. The extracted ion chromatograms were then generated for each selected fragment ion; the peak areas for the protein were obtained by summing the peak areas from 10 peptides (MS1 scan) and 7 corresponding fragment ions (MS2 scan) from each peptide. PeakView computed an FDR and a score for each assigned peptide according to the chromatograph-ic and spectra components; only peptides with an FDR below 1 % were used for protein quantization. Protein quantization was calculated by adding the peak areas of the corre-sponding peptides. Integrated peak areas were exported directly to MarkerView software (Sciex, Framingham, MA, USA) for relative quantitative analysis. The export generated three files containing quantitative information about individual ions, the summed inten-sity of different ions for a particular peptide and the summed intensity of different pep-tides for a particular protein. MarkerView uses processing algorithms that accurately find chromatographic and spectral peaks directly from raw SWATH data. Data alignment via MarkerView compensates for small variations in mass and retention time values, ensur-ing that identical compounds from different samples are accurately compared to each other. A most-like ratio (MLR) normalization was performed after statistical analysis to control for possible uneven sample loss across the different samples during the sample preparation process [16,52].

  1. Authors should mention the cut off of the statistically significant change of the proteins used as p-value and fold change.

We have included the following information in 627 and 628 lines: “The cut-off of the statistically significant change of the proteins was p-value ≤0.05 and fold change of >2”.

  1. FDRs used for the protein and peptide are missing.

We thank the review for this appreciation and apologize for this issue. We have included the following information in 555-556 lines: The false discovery rate (FDR) was set to 1% for both peptides and proteins, using a non-lineal fitting method.

10.The article needs to be reviewed for syntax, punctuation and Grammar.

We have thoroughly reviewed the entire manuscript for syntax, punctuation, and grammar. We appreciate your attention to detail and believe that these revisions have improved the clarity and quality of the article.

Round 2

Reviewer 2 Report

Comments and Suggestions for Authors

The current version is better revised, and it is acceptable now.